# Design and Preparation of Avermectin Nanopesticide for Control and Prevention of Pine Wilt Disease

**DOI:** 10.3390/nano12111863

**Published:** 2022-05-30

**Authors:** Yanxue Liu, Yiwu Zhang, Xin Xin, Xueying Xu, Gehui Wang, Shangkun Gao, Luqin Qiao, Shuyan Yin, Huixiang Liu, Chunyan Jia, Weixing Shen, Li Xu, Yingchao Ji, Chenggang Zhou

**Affiliations:** 1College of Plant Protection, Shandong Agricultural University, Tai’an 271018, China; liuyouyou2018@163.com (Y.L.); zyw15163147877@163.com (Y.Z.); xinxin990609@163.com (X.X.); yyyx19980126@163.com (X.X.); wanggehui99@163.com (G.W.); skgao@sdau.edu.cn (S.G.); lqqiao@sdau.edu.cn (L.Q.); shuyany@163.com (S.Y.); hxliu@sdau.edu.cn (H.L.); 2Shandong Research Center for Forestry Harmful Biological Control Engineering and Technology, Shandong Agricultural University, Tai’an 271018, China; 3Taishan Scenery and Scenic Spot Area Management Committee, Tai’an 271000, China; jcy1231@163.com (C.J.); swxemail@163.com (W.S.); 13561772046@163.com (L.X.)

**Keywords:** pine wilt disease, nanopesticide, avermectin, adhesion and penetration, traceable monitoring

## Abstract

Pine wilt disease is a devastating forest disaster caused by *Bursaphelenchus xylophilus*, which has brought inestimable economic losses to the world’s forestry due to lack of effective prevention and control measures. In this paper, a porous structure CuBTC was designed to deliver avermectin (AM) and a control vector insect Japanese pine sawyer (JPS) of *B. xylophilus*, which can improve the biocompatibility, anti-photolysis and delivery efficacy of AM. The results illustrated the cumulative release of pH-dependent AM@CuBTC was up to 12 days (91.9%), and also effectively avoided photodegradation (pH 9.0, 120 h, retention 69.4%). From the traceable monitoring experiment, the AM@CuBTC easily penetrated the body wall of the JPS larvae and was transmitted to tissue cells though contact and diffusion. Furthermore, AM@CuBTC can effectively enhance the cytotoxicity and utilization of AM, which provides valuable research value for the application of typical plant-derived nerve agents in the prevention and control of forestry pests. AM@CuBTC as an environmentally friendly nanopesticide can efficiently deliver AM to the larval intestines where it is absorbed by the larvae. AM@CuBTC can be transmitted to the epidemic wood and dead wood at a low concentration (10 mg/L).

## 1. Introduction

Pine wilt disease (PWD) is known as pine wood nematode (PWN) disease and pine cancer, and is a devastating forest disaster caused by a major invasive species, *Bursaphelenchus xylophilus*, which is mainly distributed in the United States, Canada, Mexico, Japan, South Korea, China, Portugal and Spain [1,2,3]. In 1968, Tokushige first isolated *B. xylophilus* from dead pine trees and inoculated the *B. xylophilus* into the xylem of pine trees, causing death to *Pinus thunbergii* Parl. and *P. koraiensis* Sieb. et Zucc [4]. PWD is highly transmissible and has brought inestimable economic and ecological losses to the world’s forestry. In nature, the spread of PWN mainly depends on vector insects, which can carry PWN and transmit it to host plants to cause disease [5,6]. Japanese pine sawyer (JPS), is considered to be the main transmission vector of PWN, which spreads the nematodes by feeding on healthy wood and supplementing nutrients, and also spreads the nematodes by laying eggs on weak and dead pine trees [7,8,9]. When JPS larvae burrow and feed on pine trees, they also cause major damage to pine trees [10,11]. At present, the prevention and treatment method of PWD is mainly through chemical agent control of the vector insects [6]. During the emergence period of JPS, pyraclosporin and thiacloprid are generally used by China, Japan, and South Korea through airplanes to spray insecticides [12,13,14]. Japan as the most severely affected country, taken as an example, where the cost of preventing and controlling PWD dominated the total expenditure on forest diseases and insect pests. In 1986, the total expenditure for prevention and control was 6.403 billion yen, of which 6.042 billion yen was used for pine wood nematode disease control [15]. In addition, the timely removal of diseased trees and dead wood (including weak wood) is an important measure for the prevention and control of PWD. There are many kinds of treatments for epidemic wood in our country: fumigation, slicing and crushing treatment and charcoal burning [16]. The cutting residue parasitized by JPS becomes an important intermediate link in the spread and epidemic of PWN because some of it contains pine wood nematodes. The scientific treatment of the residue is an important link and measure to restrain the spread and epidemic of pine wood nematodes. Although great efforts have been invested in the protection of pine wood, according to existing reports, the outlook for the control of PWD is still not optimistic. It is urgent to integrate modern science and technology to establish green prevention and control technologies for new biological agents, improve the environmental friendliness and sustainability of chemical prevention and control technologies, and build a forest ecological barrier to resist and intercept systems.

In recently years, nanotechnology has gradually been applied in agriculture and forestry [17,18]. Nanotechnology, especially, shows excellent properties in the transportation of agrochemicals, including fertilizers, pesticides and hormones [19,20,21,22]. In 2019, the International Union of Pure and Applied Chemistry (IUPAC) ranked nanopesticides as one of the top ten emerging chemical technologies that could change the world [23]. By using the special properties of nanomaterials, the defects of traditional pesticides can be resolved in a targeted manner [24,25,26,27,28]. For example, Shen used a novel carboxyl functionalized fluorescent star poly (amino acid) to deliver Bt toxin, which could reduce the resistance of pests [29]. Li used lignin to encapsulate pesticides that are easily photolyzed, which effectively prolonged DT50 by 7.89 times. However, uncontrollable nanosize and strong heterogeneity will limit the practical application of nanopesticides.

MOFs are a kind of organic-inorganic hybrid material, also called coordination polymers, which are different from inorganic porous materials and general organic complexes, [30,31,32], which have both the rigidity of inorganic materials and the flexibility of organic materials, and present huge development potential and attractive development prospects in modern materials research. MOFs exhibit high framework flexibility and shrinkage/expansion due to interactions with guest molecules [33,34]. The low volume fraction occupied by solid matter in MOFs gives them the highest porosity and surface area to date, and the coordination polymers of copper and trimesic acid (CuBTC) were reported in previous pioneering work [35]. This electrically neutral framework consists of dimeric copper tricarboxylate units with short Cu-Cu internuclear separations. Each metal completes its quasi-octahedral coordination sphere through axial water ligands opposite to the Cu-Cu vector, forming a porous structure upon removal of these water ligands from the framework [34]. In this paper, CuBTC was utilized to encapsulate the avermectin (AM) to control JPS, a major vector insect of PWN, including toxicity mechanism research, traceable pesticide monitoring and evaluation of environment, transmission efficiency, and dead wood treatment.

## 2. Materials and Methods

### 2.1. The Materials and Characterization

Avermectin (97%) was obtained from Beijing Aitemon Co., Ltd. (Beijing, China); The copper nitrate trihydrate (Cu(NO_3_) 3H_2_O, 99%), trimesic acid (H_3_BTC, 98%) and polyvinylpyrrolidone (PVP) were provided by Shanghai Macklin Biochemical Co., Ltd. (Shanghai, China); DMEM medium and fetal bovine serum (FBS) were obtained from ThermoFisher scientific Co., Ltd. (Shanghai, China); CCK-8 kit was obtained from New Cell & Molecular Bio-Technology Co., Ltd. (Shanghai, China); fluorescein isothiocyanate (FITC, >95%) and trypan blue were supplied by Coolaber Science & Technology Co., Ltd. (Beijing, China); DAPI Staining Solution and 4% Paraformaldehyde Fixative Solution were supplied by Shanghai Beyotime Bio-Technology Co., Ltd. (Shanghai, China); DF-1 cells were C/E chicken embryo fibroblasts which are resistant to endogenous ALV in the E subgroup, and were a gift from the Institute of Poultry Diseases and Tumors of the United States Department of Agriculture that have been preserved by our laboratory for generations.

X-ray diffraction (XRD) patterns were recorded with a Bruker D8 Discover Advance diffractometer using nickel-filtered Cu Kα radiation (λ = 1.5406 Å). UV–visible spectra were recorded on a DaoJin UV-2550 spectrophotometer (Tai’an, Shandong, China). Thermogravimetric analysis (TGA) was performed on a TG-DTA6300 (Beijing, China). Transmission electron microscopy (TEM) images were taken by a JEOL JEM 2100F (Beijing, China) at an accelerating voltage of 200 kV. Scanning electron microscopy (SEM) images were obtained by a JEOL JSM-6700F field-emission SEM (Beijing, China) with an accelerating voltage of 10 kV. Nitrogen sorption tests were performed in a MicroActive ASAP 2460 (2.02 version, Tai’an, Shandong, China) adsorption apparatus at 77 K up to 1 bar to obtain pore volume and pore size by analyzing nitrogen adsorption and desorption isotherms. Confocal laser scanning microscope (CLSM) images were performed on a Leica TCS SP8 microscope (Tai’an, Shandong, China). Fluorescent inverted microscope (Tai’an, Shandong, China), flow cytometer (Tai’an, Shandong, China), stereo light microscope (SLM, Nikon, SMZ25, Tai’an, Shandong, China), biological microscope (Yoke, XSP-8CA, Tai’an, Shandong, China), high performance liquid chromatography (HPLC) was configured with Waters 1525 column oven and Waters 2998 UV detector. Cell viability was measured on a ThermoFisher Varioskan Lux Multifunctional microplate reader (Tai’an, Shandong, China).

### 2.2. Preparation of AM@CuBTC

CuBTC, as an important member of the MOFs, is widely used in biomedicine, gas separation and catalysis because of its almost non-toxicity. CuBTC is composed of Cu^2+^ and the bridging organic ligand H_3_BTC through self-assembly to form a crystalline porous material with a periodic network structure. CuBTC has the characteristics of the rigidity of inorganic materials and the flexibility of organic materials that make it present great potential and attractive development prospects in application. High porosity, good chemical stability, controllable pore structure and large specific surface area of CuBTC were utilized in this paper to prepare AM@CuBTC (Figure 1). Simply, 4.5 g Cu (NO_3_) and 2 g PVP were uniformly dispersed in 250 mL methanol under 300 rpm for 30 min, and 2.15 g H_3_BTC were dissolved in 250 mL methanol under 300 rpm for 30 min. The H_3_BTC methanol solution was dropped into Cu (NO_3_) mixed solution under vigorous stirring for 30 min at room temperature. After 24 h of reaction, the complex was centrifuged at 8000 rpm for 20 min, and washed with methanol several times to obtain CuBTC precipitation, then the precipitation was replaced with deionized water twice, stored at −80 °C for 2 h before freeze drying. Next, a saturated methanol solution of AM (20 mL, 19.5 g/L) was prepared and mixed with 10 mg CuBTC under 350 rpm for 48 h. The mixture was centrifuged at 8000 rpm for 20 min and washed with ethanol several times, then the precipitation was replaced with deionized water twice, stored at −80 °C for 2 h before freeze drying to harvest AM@CuBTC powder.

### 2.3. Preparation of FITC@CuBTC

Preparation of FITC@CuBTC was similar to 2.2. Simply, FITC was mixed with deionized water, and uniform ultrasonic dispersion. CuBTC powder was dispersed into FITC solution and stirred at 300 rpm for 4 h in the dark. After treatment, the mixture was centrifuged at 14,000 rpm for 20 min to harvest FITC@CuBTC. The purification process of FITC@CuBTC was by washing with deionized water several times.

### 2.4. Pesticide Delivery

In order to study the pesticide loading capacity of CuBTC, a saturated solution of AM/methanol was prepared, 20 mg CuBTC powder was dispersed in different volumes of AM solution, and stirred at 300 rpm for 48 h. After that, the supernatant was collected at 14,000 rpm for 20 min to measure the content of free AM using a UV spectrophotometer at a detection wavelength of 245 nm. The loading efficiency was calculated according to the following formula [36].
(1)$$\mathrm{Loading}\;\mathrm{efficiency}=\left(\frac{\mathrm{Weight}\;\mathrm{of}\;\mathrm{total}\;\mathrm{AM}-\mathrm{Weight}\;\mathrm{of}\;\mathrm{AM}}{\mathrm{AM}@\mathrm{CuBTC}}\right)\times 100\%$$

### 2.5. Pesticide Release

In this article, pH and temperature-sensitive pesticide release was studied according to the method described in the literature [25]. AM@CuBTC was dispersed in PBS, transferred to a dialysis bag after being ultrasonically dispersed, and the dialysis bag put into the deionized water, then placed in a shaking box at 150 rpm for varying times. At different time points, the precipitate in the dialysis bag was collected by centrifugation, eluted with ethanol ultrasonic three times, and centrifuged to collect the supernatant, and then the release amount of AM in AM@CuBTC was detected by high performance liquid chromatography (HPLC) after filtration with 0.22 µm filter membrane. Mobile phases were methanol (A):acetonitrile (B):water (C) = 38:38:24 (*v*/*v*/*v*) with a flow rate 1.0 mL/min, injection volume 5 µL, column temperature: room temperature. The maximum absorption wavelength of AM was 245 nm.

### 2.6. Photodegradability of AM@CuBTC

A sample of 20 mg AM@CuBTC powder was dispersed in 10 mL of PBS with pH of 5.0, 7.0 and 9.0, and ultrasonically spread in a 9.0 cm petri dish, then naturally dried in the dark. The petri dishes were exposed to ultraviolet light for varying times and pH, Free AM and AM emulsifiable solution (AM@ES) were as controls. Free AM of equal quality was dissolved in methanol and also spread in 9.0 cm a petri dish, and AM@ES of equal quality was directly spread in 9.0 cm a petri dish. The anti-photolysis performance of AM@CuBTC was measured by UV spectrophotometer at a detection wavelength of 245 nm.

### 2.7. Biodistribution

Studying the fluorescence distribution of the pesticide on the surface of the JPS larva was suitable for the convenient observation of the adhesion and penetration ability of the nanoparticles. FITC@CuBTC was dispersed in deionized water and sprayed on the well-developed larvae after starving for 24 h to observe the pesticide intake of the larvae’s abdomen. The larvae were repeatedly rinsed with deionized water to remove unbound FITC@CuBTC, and then paralyzed with ethyl acetate. The main respiratory system of most insect larvae is through larval stigma, and a small amount spread through the body wall, and the fluorescence distribution in the larva was observed through stereo light microscope (SLM). In order to study the delivery process of nanoparticles in the digestive tract, the fluorescence distribution in the digestive tract of JPS larvae needs further testing. Whether AM@CuBTC can achieve effective delivery into the digestive tract will be used to evaluate the strong evidence that nanopesticides are different from traditional formulations. FITC@CuBTC solution was mixed with the artificial diet to feed to JPS larvae after starving for 24 h. After being fed for 1 h, JPS larvae were immediately made into paraffin sections, and the fluorescence distribution in the larvae was observed using SLM. JPS larvae intestines were dissected to make paraffin sections, and the cross section of the intestinal tract was detected with a fluorescence microscope. Furthermore, the epidermis of the larvae was dissected to observe the distribution of AM@CuBTC on the epidermis through SEM.

### 2.8. Cellular Uptake

Based on the fluorescence distribution of FITC@CuBTC on the body surface and digestive tract, a theoretical basis for the study of the toxicity of JPS larvae was provided. However, the specific mechanism could not be completely determined from this phenomenon, so further research at the cell level is needed. Cellular uptake of AM@CuBTC was first studied in this paper. DF-1 cells (C/E chicken embryo fibroblasts) were obtained as a model cell line, FITC@CuBTC was used to indirectly characterize the distribution of AM@CuBTC on the surface of DF-1 cells. Firstly, DF-1 cells were inoculated into a 24-well plate at a concentration of 5 × 10^4^ cells/well at 37 °C for 24 h, then transfected with AM@CuBTC, and PBS was as a control group. The cellular uptake of FITC@CuBTC was observed through CLSM.

### 2.9. Cytotoxicity of AM@CuBTC

DF-1 cells were inoculated into 96-well plates at a density of 5 × 10^3^ cells/well at 37 °C for 24 h. After incubating, the original medium of DF-1 cells was removed and transfected with fresh medium containing AM@CuBTC, and then PBS, AM@ES and free AM were as control groups. Except for the PBS group, AM@ES and AM@CuBTC had the same amount of free AM. After 4 h of transfection, the medium of DF-1 cells was changed to the original medium to continue cultivation for 96 h. The medium was removed and 25 µL, 5 mg/mL of CCK-8 added to incubate for 4 h in the dark. The supernatant was removed and 100 µL DMSO added to detect the absorbance at 245 nm by a microplate reader. The calculation formula for cell viability is as follows: [(A_Sample_ − A_Blank_)/(A_Control_ − A_Blank_)] × 100%(2)

Among them, A_Sample_ is the absorbance value of the sample group, A_Blank_ is the absorbance value of the blank group, and A_Control_ is the absorbance value of the PBS group.

### 2.10. Mechanism of Toxicity

The contact toxicity and stomach toxicity of AM@CuBTC were determined by the pesticide membrane contact method and the feed mixing pesticide method [36,37]. Firstly, the dispersed AM@CuBTC solution was placed in a petri dish, and the filter paper was slowly spread into it to make it completely soaked, then the filter paper was turned upside down to the top, and the well-developed larva after starving for 24 h were placed on the filter paper to observe the pesticide intake of the larvae’s abdomen. The well-developed 3rd-instar larvae were picked and placed on the filter paper, allowing them to crawl so that the larval epidermis was completely in contact with AM@CuBTC solution. Then the treated JPS larvae were put in an artificial breeding box of 6 cm^3^. There were 30 larvae under each concentration gradient, and 10 larvae made up a set of repeated group; the control group was treated with water.

Feed mixing pesticide method: 3 g of artificial feed were put into the breeding box, and 1 mL AM@CuBTC solution was added. The well-developed 3rd-instar larvae were put into the breeding box. There were 30 larvae under each concentration gradient, and 10 larvae made up a set of repeated group; the control group was treated with water. The breeding boxes were placed in a constant temperature incubator with a temperature of 26 ± 1 °C and a relative humidity (RH) of 65 ± 10%. The survival rate of JPS larvae was counted at 24 h and 48 h after feeding, if the larvae did not move when touched with a brush. CuBTC, AM@ES and free AM were as control.

### 2.11. Transmission of AM@CuBTC

Although it has been clarified that AM@CuBTC could effectively kill JPS larvae through the direct application process, it is still a big problem that cannot be ignored as to whether it can effectively penetrate the bark of the diseased pine into the trunk and spread through the water transportation system. After all, realizing the practical application of AM@CuBTC is our core value for preparing the nanopesticides. As an important evaluation standard, the transmission of AM@CuBTC in tree trunks was studied in this paper. The experiment base for studying the transmission of AM@CuBTC was located in Culai Mountain Forest Area in the city of Tai’an, Shandong Province, China (117.24° E, 36.05° N). An electric drill was used to drill a hole with a diameter of 5 mm and a depth of 4 cm into the trunk at a 45-degree angle downward from the ground at 30 cm. First, FITC@CuBTC was used to inject trees and the diseased pine was sawn off after varying processing times. The diseased pine was cut into small wooden pieces along the direction of the pine pith to observe the transmission of FITC@CuBTC through the small animal image. Second, AM@CuBTC was injected at concentrations of 1.0 mg/L, 10 mg/L and 50 mg/L, and AM@ES with the same concentration as a control. After 3 days, sawdust was taken 5 cm above the punch at different depths. The method of subcritical water was used to extract AM from wood chips. Simply, the sawdust was mixed with deionized water and put into a small polytetrafluoroethylene tank, then tanked into the reaction kettle and placed in a high-temperature furnace at 120 °C for 30 min. The reaction kettle was taken out, and quickly cooled, centrifuged to remove the aqueous solution. Sawdust was added to methanol and mechanically shaken for 2 min, then centrifuged to retain the supernatant, the above operation was repeated twice. The methanol solution was collected to measure the content of AM by an ultraviolet spectrophotometer. Third, the dead wood was sprayed with the concentrations of 1.0 mg/L, 10 mg/L and 50 mg/L, and AM@ES with the same concentration as a control. After 3 days, sawdust was taken 5 cm above the punch at different depths. The method of subcritical water was also used to extract AM from wood chips with the same operation method.

### 2.12. Statistical Analysis

All experiments in this work were repeated three times, and statistical analysis of the data was performed by analysis of variance (ANOVA). All the data were subjected to normality and homogeneity tests using DPS v 7.05. All graphical data are reported as the mean ± standard deviation (SD). Significance levels were set at * *p* < 0.05.

## 3. Results and Discussion

### 3.1. Characterization of AM@CuBTC

Figure 2A shows the XRD of CuBTC and AM@CuBTC; it can be seen from the X-ray diffraction pattern that the peak positions of the characteristic peaks of AM@CuBTC were basically consistent with those of the CuBTC curve, indicating that the purity and crystallinity of the samples obtained were close to CuBTC, and free AM did not change the structure of CuBTC during the process of adsorption. From Figure 2B, CuBTC had excellent adsorption properties because of its open unsaturated copper metal sites, which brought opportunities to absorb the AM. After loading the AM, the porosity of CuBTC decreased, indicating that AM occupied most of the pores. The pore size distribution showed that CuBTC was a microporous structure between 0–2 nm to 3–4 nm (Figure 2C). The changes in thermal weight loss of AM@CuBTC were observed through the thermal weight loss curve (Figure 2D). The quality of AM@CuBTC reduced by 19% in the heating range of 0–300 °C, which may be due to the decomposition of AM. Figure 3A–D shows the TEM and SEM images of AM@CuBTC and a single enlarged image, the dynamic diameter of AM@CuBTC was about 350 nm and of uniform particle size (Figure 3E).

### 3.2. Pesticide Loading of AM@CuBTC

The calibration curve of AM was linear in the concentration range of 4–64 μg/mL, the regression equation was y = 0.0115x + 0.0241, and the correlation coefficient was 0.9995 (Figure 3F). Generally speaking, the pesticide-loading capacity directly affects the field application dose of pesticides [37,38]. CuBTC powder was dispersed in different volumes of AM saturated solution to study the pesticide loading of AM@CuBTC. The loading efficiency gradually increased with the enhanced concentrations of AM, and the loading efficiency reached 40% (Figure 3F). However, the pesticide loading rate was unchanged after continuing to increase the concentration of AM, suggesting that CuBTC had reached exhaustion adsorption.

### 3.3. Assessment of Release Characteristics

Release performance of pesticides can greatly affect the delivery of pesticides in the field. Extending the slow release of a pesticide can reduce the frequency of pesticide application, and weaken large-scale environmental pollution caused by one-time release, and then facilitate long-term effective prevention and control of the target organisms [39]. This study simulated the soil environment to explore the release performance of AM@CuBTC. Under the conditions of 25 °C and pH 7.0, the release process of AM@CuBTC was stable without burst release, and the release process lasted for 72 h, the cumulative release amount reached 91.5%. In contrast, the cumulative release amount of AM@EC treatment group released 88% within 6 h, and the burst release was obvious (Figure 4A). Compared to the commercial formulations, the release performance of AM@CuBTC was better; it had a longer sustained release time and higher cumulative release amount. AM@CuBTC released stably at different pH, and the release rate was higher than that of pH 5.0 and 9.0. The cumulative release amount of pH 5.0 and 9.0 was 83.5 and 79.2%, respectively. To achieve the desired control effect in agriculture and forestry, the pesticides were often overused due to the short release duration, which increased production costs and exacerbated the problem of residues and pesticide resistance [40,41]. Hence the development of AM@CuBTC could give a new chance for making full use of the effects of pesticides and continuing to administer them. Next, the release of AM@CuBTC under different temperature conditions was explored (Figure 4B). Compared with the current traditional AM@ES, the release of AM@CuBTC remained stable with cumulative release exceeding 90% after 96 h. With the increase of temperature, the release rate of AM gradually improved (91.4% at 35 °C). Obviously, the development of AM@CuBTC is of great significance for reducing the pressure of agriculture on the environment and reducing the amount of traditional pesticides [42,43].

### 3.4. Photodegradability of AM@CuBTC

The structure of the conjugated double bond and the 16-membered ring macrolide structure made AM easy to degrade under light [44,45]. So, it is necessary to consider the anti-photolysis effect of CuBTC on the AM when preparing AM@CuBTC. The retention curve of AM exposed to ultraviolet light clarified the protective effect of AM@CuBTC on AM (Figure 5). By comparison, it can be observed that the retention rate of each treatment group decreased with the extended treatment time under the condition of ultraviolet light irradiation. After 72 h, the retention rate of the free AM was less than 5%, and the AM@ES treatment group was only about 15%, while the AM@CuBTC treatment group was still more than 70%. The possible explanation is that as AM@CuBTC gradually released AM after treatment, the released AM was degraded, resulting in a decrease in retention rate, indicating that CuBTC could effectively protect AM from photolysis. In addition, the retention rates of the AM@CuBTC at pH 5.0 and 9.0 after 120 h were 62.2 and 69.4%, respectively. Obviously, the weak degradation of CuBTC under acidic conditions promoted the release of AM with the prolongation of the treatment time, and increased the photolysis rate, and then reduced the retention rate of AM. The excellent anti-photolysis performance of AM@CuBTC was mainly attributed to its porous structure, and AM was absorbed on the pores inside CuBTC during the preparation of AM@CuBTC, which naturally shielded it from exposure to ultraviolet light.

### 3.5. Biodistribution

In order to further verify the biodistribution of AM@CuBTC to JPS larvae, FITC-labeled CuBTC was used for the fluorescence test. Fluorescent microscope and SEM were used to explore the coverage, permeability and adhesion of AM@CuBTC on the larval epidermis. It was clearly observed that the morphology of JPS larvae was successfully labeled with fluorescence under the dark field excitation light of 488 nm, unlabeled CuBTC-treated JPS larvae served as controls (Figure 6A,B). There was fluorescence distributed on the entire epidermal structure of JPS larvae, especially in the folds, where the fluorescence was intensified, meaning that CuBTC has high adhesion properties. After crawling for a period of time, the fluorescence did not fall off, indicating that FITC@CuBTC had good adhesion on the larval epidermis. There were a large number of fluorescent particles distributed around the valve on both sides of the larvae. As we all know, the respiration of insect larvae mainly depends on the larval stigma, followed by the body wall, [46,47]. so we enlarged the larval stigma of the JPS larva to observe whether there was fluorescence distribution (Figure 6C,D). As we expected, the larval stigma of the JPS larva was enriched with fluorescence and spread to the depths when JPS larva breathed followed by the valve stigma opening and closing. We speculated that these larval stigmas would be the main channel for the FITC@CuBTC to enter the bodies of the larvae. It also confirmed that FITC@CuBTC was of good permeability and passed through the epidermal structure in a very short time, implying that AM@CuBTC could effectively achieve contact toxicity.

In addition, through observing the cross-sectional fluorescence distribution of the JPS larva, the intestinal inside was clearly observed from Figure 6E, revealing the fact that FITC@CuBTC was delivered from the body wall to the body tissue, also meaning that CuBTC could be transferred to subcutaneous tissue through the body wall from the epidermis to the intestine and be enriched in the body. In order to further observe the distribution of fluorescence inside the intestine, the cross-section of different parts of JPS larvae intestine were sliced and dyed with HE, and then observed through a fluorescence microscope. From Figure 6F,G, there was obvious fluorescence inside the intestine of JPS larvae, and the outline was clearly visible, which provided a basis for the delivery of AM. It was noticed that the intestine showed an enhanced distribution of green fluorescence, which indicated that pesticide delivery from the stomach to the intestine could be achieved by feeding.

### 3.6. Cellular Uptake and Detection of Adhesion

We used avian DF-1 cells as model cells and observed the uptake of FITC@CuBTC on cells through flow cytometry and CLSM. From Figure 6H, the uptake rate of cells detected by flow cytometry was as high as 90–100%. FITC@CuBTC was fully adsorbed to the cell surface and exhibited enhanced fluorescence uptake (Figure 6I,J). It was obvious that CuBTC greatly promoted the delivery and sustained release of AM in JPS larvae. Contact toxicity, namely pesticide entering the pest body through the epidermis after contact to exert its activity, is limited by the dispersibility of traditional preparations [48,49]. The poor coverage adhesion and permeability of active ingredients on the larval epidermis is an important factor for restricting contact toxicity. AM@CuBTC was embedded well into the texture structure on the larval epidermis and did not easily fall off. AM@CuBTC particles were mainly distributed on the folds of the epidermal structure and around the valve and the roots of the chaeta. Compared with traditional pesticide formulations, the smaller particle size, larger specific surface area, and higher water dispersibility of the nanopesticide make for better coverage, adhesion and permeability [50,51,52]. Because the nanoformulation significantly improved the diffusion and permeability of pesticides, the active ingredients could enter into the larval body more efficiently, finally the insecticidal activity was enhanced.

### 3.7. Cytotoxicity Assessment of AM@CuBTC

The cytotoxicity of AM@CuBTC, AM@ES, Free AM and CuBTC was measured in vitro. From Figure 7A, CuBTC had little toxicity to DF-1 cells (16 μg/mL, 18%), and the cytotoxicity of the AM@CuBTC was higher than that of AM@ES and free AM. The mortality rate of AM@CuBTC was 13.9% at 1 μg/mL, while free AM was only 6.19%, indicating that AM@CuBTC had a greater increase in cytotoxicity at a low concentration, and not caused by the cytotoxicity of CuBTC (0.95%). In addition, the effects of different temperatures on cell mortality (Figure 7B) were also examined. With the temperature increased (from 15 °C to 35 °C), the cell mortality in each treatment group gradually improved. Compared with free AM and AM@ES, AM@CuBTC had a greater impact on cell mortality. The effect of different treatment times on cell mortality was researched (Figure 7C), as the treatment time extended, the cell mortality of each treatment group gradually increased. The mortality of AM@CuBTC (60.6%) was higher than free AM (39.7%) and AM@ES (44.8%) after 8 h of treatment. In summary, AM@CuBTC can effectively enhance the cytotoxicity and utilization of AM, which will provide valuable research value for the application of typical plant-derived nerve agents in the prevention and control of forestry pests.

### 3.8. Mechanism of Toxicity

Contact toxicity refers to the pesticide entering the pest body by contacting the epidermis of the insects to exert its activity [53]. Due to the poor dispersion of traditional formulations, and the unsatisfactory coverage, permeability and adhesion, the active ingredients could not reach the site of action through the larval epidermis, resulting in unsatisfactory contact toxicity [54]. Stomach toxicity refers to the pesticide entering the body through the mouthparts of the pest and exerting its activity during the feeding process [55]. However, the limited life of the active ingredients results in failure to reach the larval intestine efficiently and to be absorbed by the pest [54]. The researched mechanism of contact toxicity and stomach toxicity of AM@CuBTC is shown in Figure 1. The enhancement of AM@CuBTC for contact toxicity and stomach toxicity was researched at the individual level, the data of which is listed in Figure 8. The corrected mortality rates of AM@CuBTC were 8.9% (5 μg/mL), 22.2% (10 μg/mL), 40.0% (20 μg/mL), 52.2% (40 μg/mL) and 66.7% (80 μg/mL), respectively, and the corrected mortality rates of AM@ES were 4.5% (5 μg/mL), 19.1% (10 μg/mL), 35.5% (20 μg/mL), 43.7% (40 μg/mL) and 52.9% (80 μg/mL), respectively, while the corrected mortality rates of free AM were 3.1% (5 μg/mL), 16.7% (10 μg/mL), 28.2% (20 μg/mL), 37.9% (40 μg/mL) and 47.8% (80 μg/mL), respectively (Figure 8A). The lethality rate of AM@CuBTC increased by 70.5% at low concentrations (5 μg/mL), while the contact toxicity increased by only 14.5% with concentrations of 80 μg/mL. In general, the AM was absorbed by CuBTC to prepare the AM@CuBTC nanopesticide that could effectively improve the contact toxicity of active ingredients to target organisms, which also confirmed the results of the previous parts of research. The stomach toxicity was higher than the contact toxicity with different concentrations (Figure 8B). The AM@CuBTC had the highest corrected mortality of 13.2% (5 μg/mL), 29.1% (10 μg/mL), 48.7% (20 μg/mL), 61.5% (40 μg/mL), and 90.4% (80 μg/mL), respectively. In general, the prepared AM@CuBTC can effectively improve the contact toxicity and stomach toxicity of AM. AM@CuBTC had a good dispersibility and efficient delivery of AM to the larval intestines and was absorbed by the larvae. After entering the larvae, AM@CuBTC could effectively increase the cell uptake rate, thereby increasing cytotoxicity.

### 3.9. Transport Efficiency

Because the outer epidermis of the tree is dead tissue, it is composed of keratinized cells, produced by cork cambium [56]. Cork is the main component of the outer layer of the bark, which can insulate moisture and gas from passing through, protect the tree, and is a natural barrier for chemical control [57]. Therefore, we deliver nanopesticides into the tree by injection. AM@CuBTC was designed to control PWN and its vector insect JPS, and the transportation efficiency in the pine tree trunk would directly affect its application prospects. As shown in Figure 9, at the same concentration, the transmission efficiency of AM@CuBTC was significantly higher than that of AM@ES.

The higher the concentration of AM@CuBTC, the higher the concentration of the pesticide reaching the pith. AM@CuBTC could be transmitted to the tree pith at a low concentration of 10 mg/L with the transport of water, while it is obviously difficult for AM@ES to reach the tree pith. Compared to tree trunks injected with water (Figure 10A), the strong fluorescence distribution was observed by peeling off the bark (Figure 10B), and fluorescence intensity decreased gradually along the pith direction (Figure 10C). However, with the extension of the delivery time, we unexpectedly found that the fluorescence at the pith began to be enriched (Figure 10D).

What is the reason for this phenomenon? We speculated that the rate of water transport along the pith was different, the closer to the pith tree, the faster the water transport, and the greater the pesticide concentration difference between inside and outside, which led to the continuous transport of FITC@CuBTC to the pith (Figure 10). This would also inhibit the irreversible damage caused by the migration of nematodes to the middle of the tree, indicating that AM@CuBTC would effectively protect the main part of the tree from pests. The treatment of dead wood was also very important during transportation. [56,58,59]. In this article, AM@CuBTC was injected into dead wood and sprayed AM@CuBTC (adhesion of nanoparticles) on the bark surface to seal the dead wood. It can be seen from Figure 11 that AM@CuBTC could also be transmitted in dead wood. However, the transmission efficiency in dead wood at the same concentration was much lower than that of epidemic wood, and the same was true for AM@ES transmission. This phenomenon can be understood as the basic stagnation of the water transport in dead wood, the delivery of AM@CuBTC is blocked, and efficient diffusion cannot be achieved in dead wood; the only water remaining in the dead wood provided effective support for the delivery of AM@CuBTC, which was also based on the good diffusion performance of CuBTC to promote the further transmission of AM@CuBTC.

## 4. Conclusions

We first proposed a design to load avermectin through CuBTC for the prevention and control of the pine wood nematode and using the vector insect, the Japanese pine sawyer, including an evaluation of the toxicity mechanism and traceable pesticide monitoring. We utilized the high porosity and surface area of CuBTC (coordination polymers of copper and trimesic acid) that was chosen as the biocompatible material for the delivery of avermectin to improve the solubility, photolysis performance and pesticide efficacy. From the traceable monitoring experiment, AM@CuBTC easily penetrated the body wall of the Japanese pine sawyer larvae and was transmitted to tissue cells through contact and diffusion. Furthermore, AM@CuBTC can effectively enhance the cytotoxicity and utilization of AM, which will provide valuable research value for the application of typical plant-derived nerve agents in the prevention and control of forestry pests. Moreover, AM@CuBTC could be transmitted to the epidemic wood and dead wood at a low concentration. In the optimal control period of the longhorn beetle larvae, nanodrugs are injected into the parts above the root of the tree by punching the borer, and the nanodrugs are delivered into the interior through the water transport of the tree, diffusing layer by layer. With the release of the hormone, the sustained drug effect will control the larvae, egg hatching and nematode reproduction, which will effectively reduce the incidence of the final disease, as well as having an effective preventive effect, the slow release of the drug greatly improves the insecticidal efficiency.

## Figures and Tables

**Figure 1 nanomaterials-12-01863-f001:**
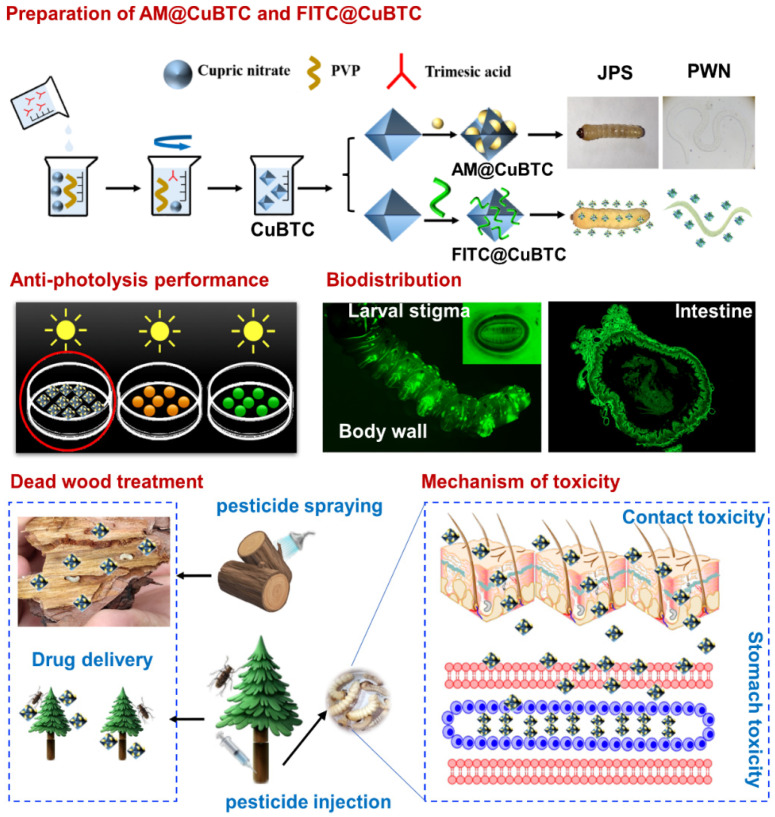
Schematic illustration of preparing avermectin @CuBTC and FITC@CuBTC, anti-photolysis performance of free avermectin, avermectin @ES and avermectin @CuBTC, biodistribution of FITC@CuBTC and traceable pesticide monitoring, epidemic wood transmission of avermectin @CuBTC, plague tree treatment, and mechanism of toxicity (contact toxicity and stomach toxicity).

**Figure 2 nanomaterials-12-01863-f002:**
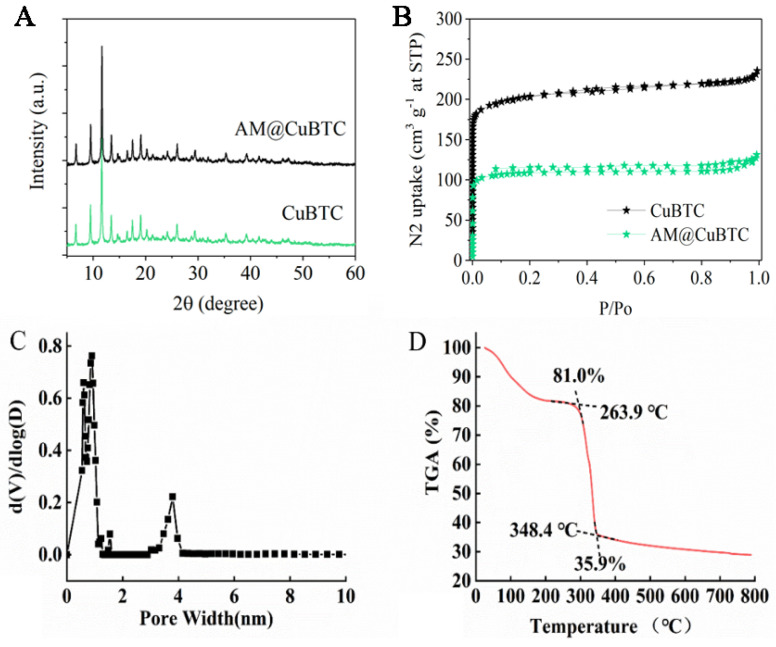
The XRD and BET of CuBTC and AM@CuBTC (**A**,**B**); mesoporous distribution of CuBTC (**C**), and TGA (**D**) of CuBTC.

**Figure 3 nanomaterials-12-01863-f003:**
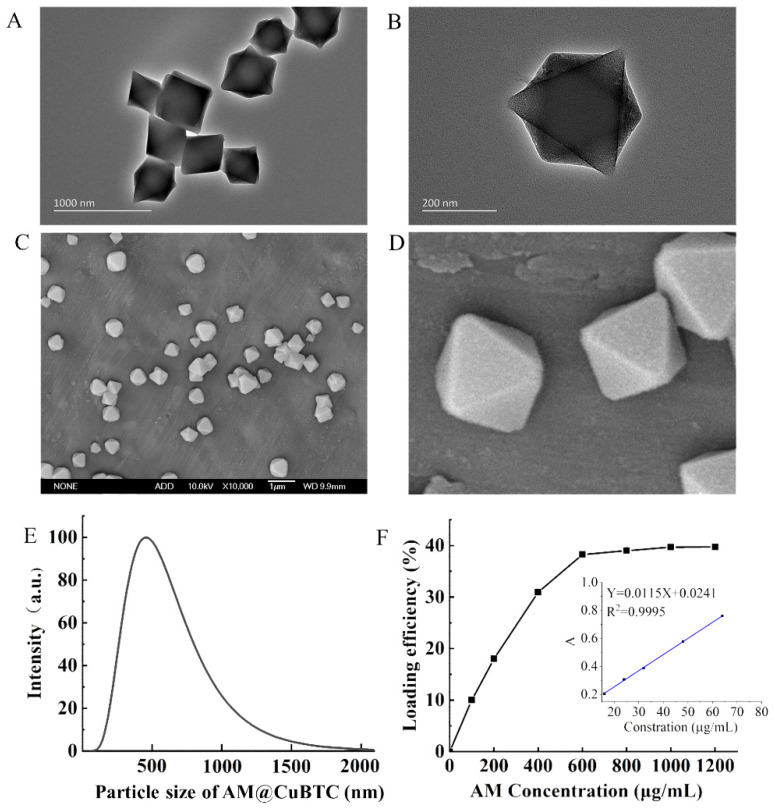
The TEM image of avermectin @CuBTC (**A**), and magnified image (**B**); the SEM image of avermectin @CuBTC (**C**), and magnified image (**D**). The particle size distribution of avermectin @CuBTC (**E**); the loading efficiency of avermectin @CuBTC and the standard curve of avermectin (**F**).

**Figure 4 nanomaterials-12-01863-f004:**
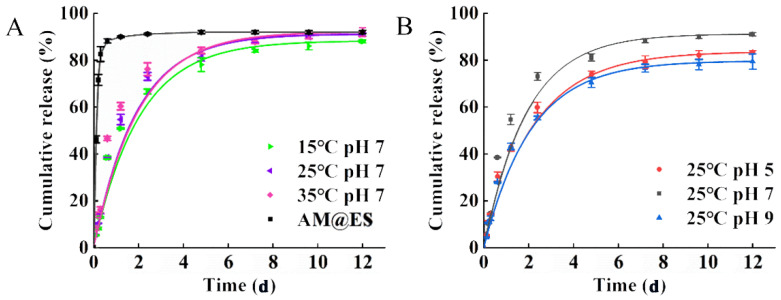
(**A**) The release curve of avermectin @CuBTC at 15 °C, 25 °C and 35 °C with pH of 7.0, and avermectin @ES as a control; (**B**) the release curve of avermectin @CuBTC with pH of 5.0, 7.0 and 9.0 at the temperature of 25 °C.

**Figure 5 nanomaterials-12-01863-f005:**
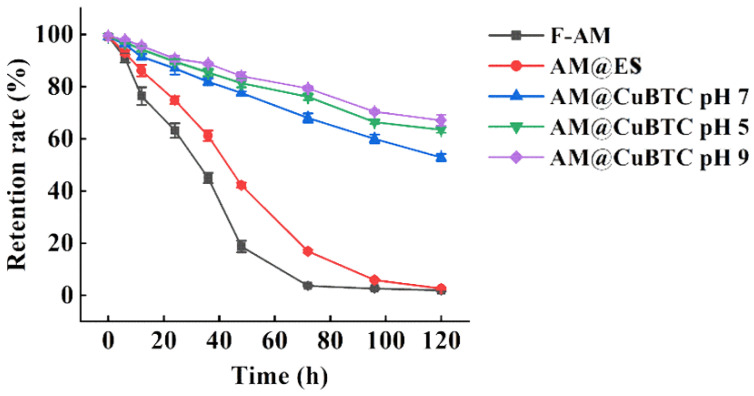
The retention rate of free avermectin, avermectin @ES, avermectin @CuBTC with pH of 5.0, 7.0 and 9.0 after being exposed to ultraviolet light (emitted by a 30 W, 310 nm lamp).

**Figure 6 nanomaterials-12-01863-f006:**
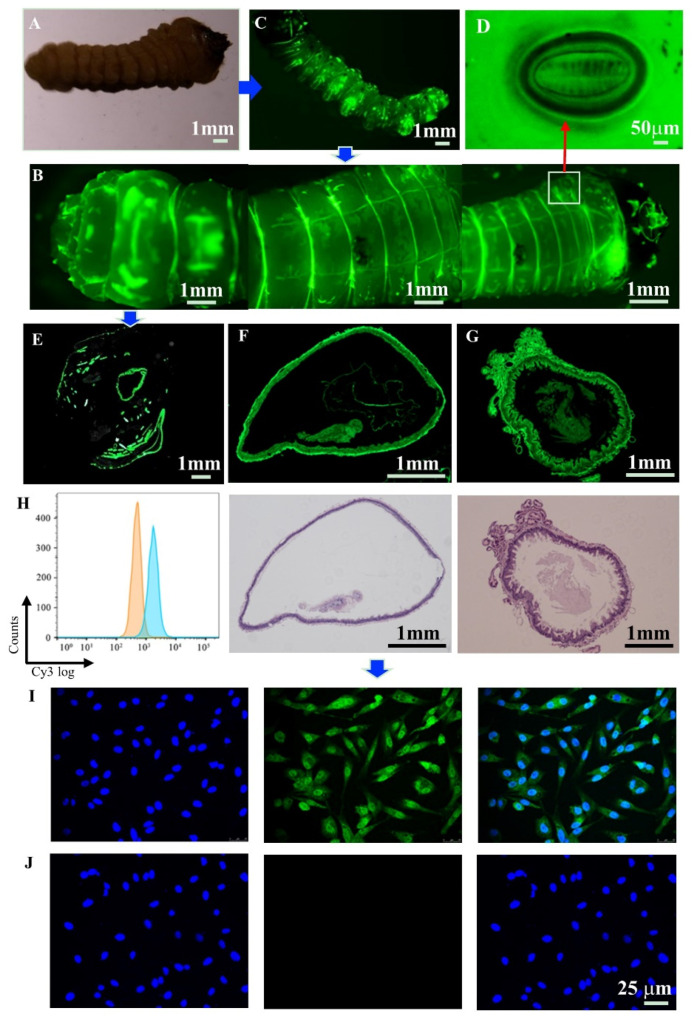
Fluorescence distribution of JPS larva after treatment with FITC@CuBTC (**B**), JPS larva without fluorescence treatment as a control (**A**); abdominal fluorescence distribution of JPS larvae (**C**); magnified image of larval stigma (**D**); (**E**) fluorescence distribution of the middle cross section of JPS larva; (**F**) fluorescence distribution image of the midgut cross section and HE staining; (**G**) fluorescence distribution image of foregut cross section and HE staining; (**H**) the uptake rate of FITC@CuBTC was detected by flow cytometry. (**I**) The cellular uptake of FITC@CuBTC to DF-1 cells (green), DAPI stains the nucleus (blue), and merge image; (**J**) the cellular uptake of CuBTC to DF-1 cells (green), DAPI stains the nucleus (blue), and merge image.

**Figure 7 nanomaterials-12-01863-f007:**
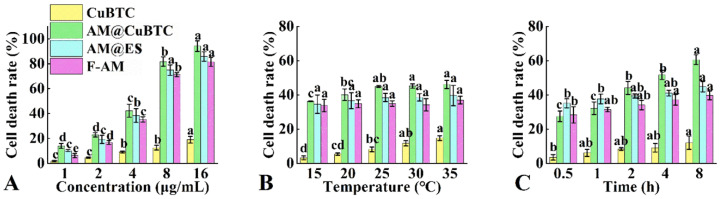
The cell death rate of CuBTC, avermectin @CuBTC, avermectin @ES, and free avermectin with different concentrations (**A**), temperatures (**B**), and treatment times (**C**). Data in the figure mean ± SD of three replications (*n* = 3), different letters indicate significant differences (ANOVA, *p* < 0.05).

**Figure 8 nanomaterials-12-01863-f008:**
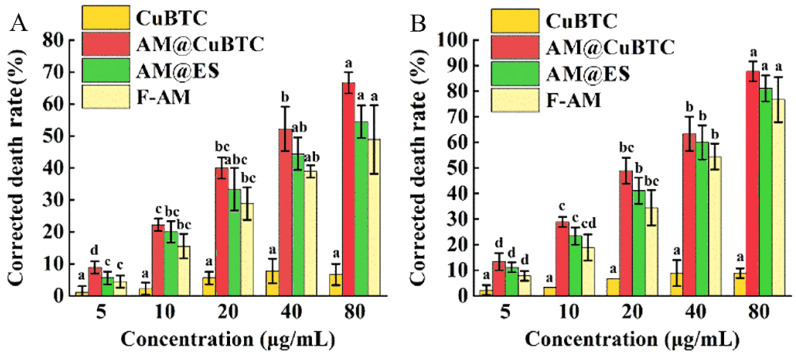
The contact toxicity of CuBTC, free avermectin, avermectin @ES and avermectin @CuBTC at 48 h (**A**), and the stomach toxicity at 48 h (**B**). Data in the figure are mean ± SD of three replications (*n* = 3), different letters indicate significant differences (ANOVA, *p* < 0.05).

**Figure 9 nanomaterials-12-01863-f009:**
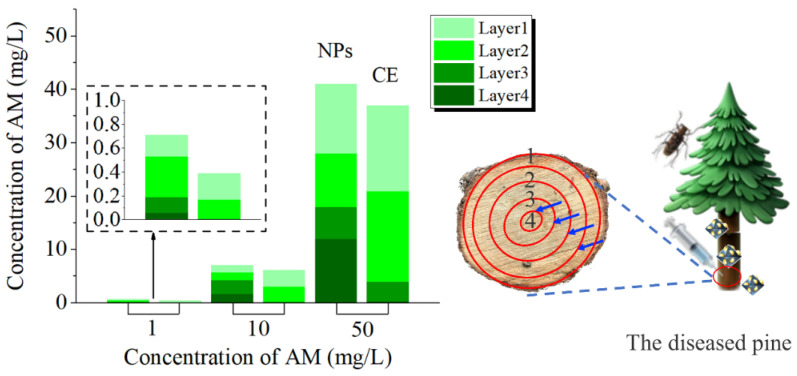
The content of avermectin in the 1–4 layers of the pine trunk changes after injecting with avermectin @CuBTC, and avermectin @ES as a control.

**Figure 10 nanomaterials-12-01863-f010:**
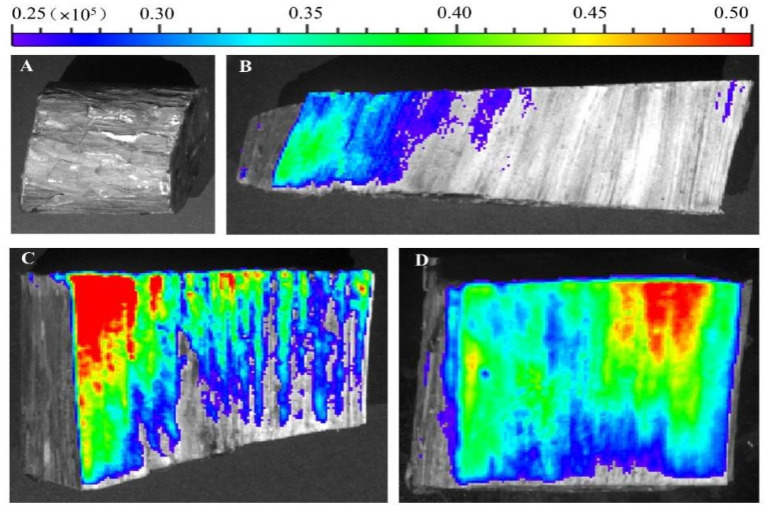
Tree trunks injected with water (**A**), fluorescence distribution along the pith direction after 1 h (**B**), fluorescence intensity along the pith direction after 5 h (**C**), fluorescence intensity along the pith direction after 24 h (**D**).

**Figure 11 nanomaterials-12-01863-f011:**
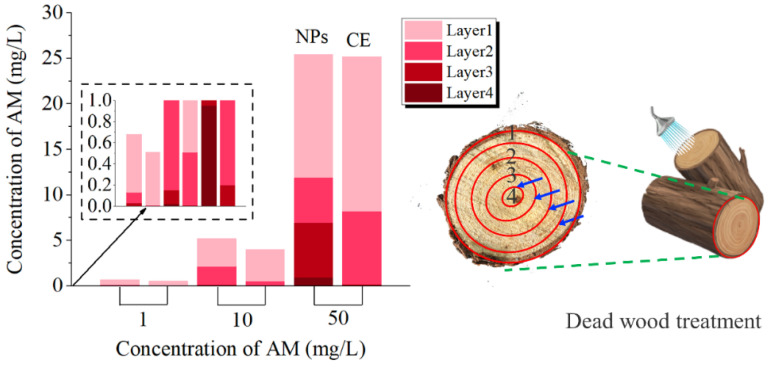
The content of avermectin in the 1–4 layers of the dead wood after injecting with avermectin @CuBTC, and avermectin @ES as a control.

## Data Availability

The data are included in the article.

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
