# Peer review of "Design and Preparation of Avermectin Nanopesticide for Control and Prevention of Pine Wilt Disease"

_nanomaterials, 2022, doi:10.3390/nano12111863_

Round 1
Reviewer 1 Report
nanomaterials-1675916
This paper report High Adhesion and Penetration of Avermectin@CuBTC Nano-Pesticides for Prevention and Control Pine Wilt Disease, and in vivo Traceable Monitoring. The research was done systematically and very comprehensively. Minor comments below can be addressed to enhance the manuscript quality.
There seems lack of literature survey on CuBT, let alone for pesticide application. The rationale behind the material selection and important properties of the material needs to be discussed to govern a solid argument.
Line 504: Was there any attempt to prove the speculation on the water diffusion mechanism? The speculation is very interesting and can unravel the fundamental transport mechanism of water in dead and living trees.
Please rewrite the sentence in lines: 15-19. It is too long and hard to read.
Few typos are still found in the manuscript.
Author Response
Response to Reviewer 1 Comments
This paper report High Adhesion and Penetration of Avermectin@CuBTC Nano-Pesticides for Prevention and Control Pine Wilt Disease, and in vivo Traceable Monitoring. The research was done systematically and very comprehensively. Minor comments below can be addressed to enhance the manuscript quality.
Point 1: There seems lack of literature survey on CuBTC, let alone for pesticide application. The rationale behind the material selection and important properties of the material needs to be discussed to govern a solid argument.
Response 1: Thanks for your opinions and encouragements for our papers. I have added literature survey on CuBTC and made the following corrections.
MOFs are a kind of organic-inorganic hybrid materials, also called coordination polymers, which are different from inorganic porous materials and general organic complexes, [27-29] which has both the rigidity of inorganic materials and the flexibility of organic materials, and presents huge development potential and attractive development prospects in modern materials research. MOFs exhibit high framework flexibility and shrinkage/expansion due to interactions with guest molecules. [30, 31] The low volume fraction occupied by solid matter in MOFs gives them the highest porosity and surface area to date, and coordination polymers of copper and trimesic acid (CuBTC) were reported in previous pioneering work. [32] This electrically neutral framework consists of dimeric copper tricarboxylate units with short Cu-Cu internuclear separations. Each metal completes its quasi-octahedral coordination sphere through axial water ligands opposite to the Cu-Cu vector, forming a porous structure upon removal of these waters from the framework. [31]
- Greathouse, J. A.; Allendorf, M. D. The interaction of water with MOF-5 simulated by molecular dynamics. J. Am. Chem. Soc. 2006, 128, 10678-10679.
- Gould, S. L.; Tranchemontagne, D.; Yaghi, O. M.; Garcia-Garibay, M. A. Amphidynamic character of crystalline MOF-5: rotational dynamics of terephthalate phenylenes in a free-volume, sterically unhindered environment. J. Am. Chem. Soc. 2008, 130, 3246-3247.
- Yao, C.; Lykourinou, V.; Vetromile, C.; Hoang, T.; Ming, L. J.; Larsen, R. W.; Ma, S. Q. How can proteins enter the interior of a MOF? Investigation of cytochrome translocation into a MOF consisting of mesoporous cages with microporous windows. J. Am. Chem. Soc. 2012, 134, 13188-13191.
- Pan, T.; Shen, Y.; Wu, P.; Gu, Z. D.; Zheng, B.; Wu, J. S.; Li, S.; Fu, Y.; Zhang, W. N.; Huo, F. W. Thermal Shrinkage Behavior of Metal–Organic Frameworks. Adv. Funct. Mater., 2020, 30, 2001389.
- Kang, W.; Zhang, Y.; Fan, L.; Dai, F.; Wang, R.; Sun, D. Metal-Organic Framework Derived Porous Hollow Co3O4/N-C Polyhedron Composite with Excellent Energy Storage Capability. ACS Appl. Mater. Inter., 2017, 9(12): 10602–10609.
- Rosi, N. L.; Eddaoudi, M.; Kim, J.; O'Keeffe, M.; Yaghi, O. M. Advances in the chemistry of metal–organic frameworks. Crystengcomm, 2002, 4: 401-404.
Point 2: Line 504: Was there any attempt to prove the speculation on the water diffusion mechanism? The speculation is very interesting and can unravel the fundamental transport mechanism of water in dead and living trees.
Response 2: Thanks for your advice, the speculation of water diffusion mechanism in dead and living trees is very interesting indeed. That's part of what we're doing right now, we've found in drug delivery that moisture has an effect on drug diffusion, so we're measuring healthy trees in the same area and trees with different levels of infestation, and moisture from root to top, on the one hand, for better pesticide application, and on the other hand, to explain the transport mechanism between dead and living trees.
Point 3: Please rewrite the sentence in lines: 15-19. It is too long and hard to read. Few typos are still found in the manuscript.
Response 3: Thanks for your advice, I have rewritten the sentence in lines: 15-19.
In this paper, a porous structure CuBTC was designed to delivery Avermectin (AM@CuBTC) and control vector insect Japanese pine sawyer (JPS) of B. xylophilus, which can improve biocompatibility, anti-photolysis and delivery efficacy of AM.

Reviewer 2 Report
Dear Authors:
In my viewpoint, the manuscript Number nanomaterials-1675916 titled " High Adhesion and Penetration of Avermectin@CuBTC Nano- Pesticides for Prevention and Control Pine Wilt Disease, and in vivo Traceable Monitoring ", cannot be published as ist being ranked as rejected. This ranking is justified below. I suggest that authors cancel this submission , provide further revision and upgrade of references, as well as re submit manuscript with new ID number.
With objective of an easy identification of points that necessitate some verification, I make mine observations in according to major topics.
Title:
As a whole, the title needs be re-designed. At moment, the title isn’t good but can made better. See, at least, title cannot contain any real or functional acronym. Then, symbol for functionalized nanoparticle should be cutter off.
At moment, the title exhibit mechanical problem with two types or size of letter, see “…”prevention and control”…
Also is too long. Then I suggest that authors cut off …”and in vivo Traceable Monitoring”. As a matter of fact, …”in vivo Traceable Monitoring” can be transferred to Abstract.
As mentioned previously, the functional acronym Avermectin@CuBTC does not should be used, at least in Title.
I suggest something similar to Design and preparation of functional Nano-pesticide for control and prevention of Pine wilt disease.
Abstract:
At moment, the item abstract is not functional. Seems that abstract is non-homogeneous. I suggest that Avermectin be classified in chemical terms or/and its biological action described. Also, CuBTC should be presented from industrial name or commercial name and its chemical one (IUPAC) provided, in the experimental item.
Introduction:
As a whole, the item introduction is directed to exhibit the-state-of-art of an event, technology, material, or Nano-something. At moment, seems that introduction item is incomplete. There is a complete absence of focus. Seems that the state-of-art of topics of interest isn’t approached in a due way would be nano encapsulating process, and/or its materials for nano encapsulating or functional toxins. Also, in the actual version, 29 references are cited only in the introduction item, when the total of references is equal to 37. Therefore, the total number of references is very small. I suggest that further bibliographic revision be carried out. A general model suggest that 1/3 of references be allocated in the introduction, 1/3 in Experimental and 1/3 in Discussion (Results and Discussion). As a matter of fact, 2/3 of them would be cited in the Results and Discussion, in a broad sense. I suggest that number of citations increase up to 3 times. See, the introduction should be re-written being necessary move Figure 1 to Materials and Methods.
Materials and methods:
As a whole, I felt the absence of character “Nano” or nanostructures. I suggest that the excess of acronyms prejudice the understanding the several topics. At moment, it is unclear that type of nanostructure is prepared. If is a nanoparticle or Nano capsule. It nature also is unclear if solid or another.
I suggest that Infrared spectrum and UV-Vis of Avermectin should be added to manuscript or added to Complementary file.
Results and Discussion:
As a whole, all Figures’ legend should be re-written. Figure legends shouldn’t contain acronyms.
In a general way, the manuscript should be written is an impersonal way but in the 398 line appears “we used… In this sense, an impersonal form is close to Avian DF-1 cells were used….
In Fig 2 all hkl planes should be inserted. The X ray diffraction has several diffraction lines but not peaks. Each diffraction line is indexed at one hkl plane. At moment, this figure is very small and should be increased. At moment, there is a lack of information at about adsorption of Avermectin in the surface of nanoparticle, is unclear if this compounds penetrate at surface region. I suggest that infrared at mid region be carried out at least CuBTC and AM@CuBTC.
Figure 3A, 3B, 3C e 3D are very small, and at moment is impossible read bar scale of magnification. The figure should be size increased.
The selection of verbs should be making with extreme careful. In the 455 line, see …was listed in Fig 9. See, the verb to list should be used in a Table.
Conclusion:
This item is constructed in an erroneous way. Summary is a poor listing of events picked up of Results and Discussion. Then, the item is Conclusion or Summary. If the item is conclusion a set of contributions can be listed. Seems that is viable to write a good conclusion item. See, at moment, this item can be re-written. Sound estrange …“We speculate…”, I suggest insert only one major contributions, and two or three secondary contributions.
Again, I believe that authors make use acronyms in an erroneous way. I does not recommend make use of acronyms in the conclusion item. It is possible, two times a word without problems.
Author Contributions:
I suggest that complete names and its affiliations be added.
Conceptualization, Y. X. L.; Funding acquisition, C. G. Z.; Methodology, Y. 539 W. Z., X. X., X. Y. X., G. H. W., S. K. G., L. Q. Q., S. Y. Y. and H. X. L.; Resources, C. Y. J., W. X. S. and 540 L. X.; Writing – original draft, Y. X. L.; Writing – review & editing, Y. C. J. All authors have read and 541 agreed to the published version of the manuscript.

Author Response
Response to Reviewer 2 Comments
Dear Authors:
In my viewpoint, the manuscript Number nanomaterials-1675916 titled " High Adhesion and Penetration of Avermectin@CuBTC Nano- Pesticides for Prevention and Control Pine Wilt Disease, and in vivo Traceable Monitoring ", cannot be published as list being ranked as rejected. This ranking is justified below. I suggest that authors cancel this submission, provide further revision and upgrade of references, as well as re submit manuscript with new ID number. With objective of an easy identification of points that necessitate some verification, I make mine observations in according to major topics.
Title:
Point 1: As a whole, the title needs be re-designed. At moment, the title isn’t good but can made better. See, at least, title cannot contain any real or functional acronym. Then, symbol for functionalized nanoparticle should be cutter off.
At moment, the title exhibit mechanical problem with two types or size of letter, see “…”prevention and control”…
Also is too long. Then I suggest that authors cut off …”and in vivo Traceable Monitoring”. As a matter of fact, …”in vivo Traceable Monitoring” can be transferred to Abstract.
As mentioned previously, the functional acronym Avermectin@CuBTC does not should be used, at least in Title.
I suggest something similar to Design and preparation of functional Nano-pesticide for control and prevention of Pine wilt disease.
Response 1: Thanks for your advice, I have changed title to “Design and preparation of Avermectin Nano-pesticide for control and prevention of Pine wilt disease” after careful consideration.
Abstract:
Point 2: At moment, the item abstract is not functional. Seems that abstract is non-homogeneous. I suggest that Avermectin be classified in chemical terms or/and its biological action described. Also, CuBTC should be presented from industrial name or commercial name and its chemical one (IUPAC) provided, in the experimental item.
Response 2: Thanks for your advice, based on the suggestions you mentioned, we have made corrections as follows:
Pine wilt disease is a devastating forest disaster caused by Bursaphelenchus xylophilus, which has brought inestimable economic losses to the world's forestry due to lack of effective prevention and control measures. In this paper, a porous structure coordination polymers of copper and trimesic acid (CuBTC) was designed to delivery Avermectin (AM@CuBTC) and control vector insect Japanese pine sawyer (JPS) of B. xylophilus, which can improve biocompatibility, anti-photolysis and delivery efficacy of AM. The results illustrated the cumulative release of pH-dependent AM@CuBTC was up to 12 days (91.9%), and also effectively avoid photodegradation (pH 9.0, 120 h, retention 69.4%). From traceable monitoring experiment, AM@CuBTC made it easily penetrate the body wall of JPS larvae and transmit to tissue cells though contact and diffusion. Otherwise, AM@CuBTC can effectively enhance the cytotoxicity and utilization of AM, which will provide valuable research value for the application of typical plant-derived nerve agents in the prevention and control of forestry pests. AM@CuBTC as an environmentally friendly nano-pesticide can efficiently deliverys AM to the larval intestines and absorbed by the larvae. AM@CuBTC could be transmitted to the epidemic wood and dead wood at a low concentration (10 mg/L). We speculate that AM@CuBTC will break through the existing research limitations and bring new opportunity to research forestry pests.
Introduction:
Point 3: As a whole, the item introduction is directed to exhibit the-state-of-art of an event, technology, material, or Nano-something. At moment, seems that introduction item is incomplete. There is a complete absence of focus. Seems that the state-of-art of topics of interest isn’t approached in a due way would be nano encapsulating process, and/or its materials for nano encapsulating or functional toxins. Also, in the actual version, 29 references are cited only in the introduction item, when the total of references is equal to 37. Therefore, the total number of references is very small. I suggest that further bibliographic revision be carried out. A general model suggest that 1/3 of references be allocated in the introduction, 1/3 in Experimental and 1/3 in Discussion (Results and Discussion). As a matter of fact, 2/3 of them would be cited in the Results and Discussion, in a broad sense. I suggest that number of citations increase up to 3 times. See, the introduction should be re-written being necessary move Figure 1 to Materials and Methods.
Response 3: Thanks for your advice, we have corrected the introduction section based on the suggestion you mentioned and as follows. In addition, adjusted the reference structure of the entire article, while moving Figure 1 to the Materials and Methods section.
Materials and methods:
Point 4: As a whole, I felt the absence of character “Nano” or nanostructures. I suggest that the excess of acronyms prejudice the understanding the several topics. At moment, it is unclear that type of nanostructure is prepared. If is a nanoparticle or Nano capsule. It nature also is unclear if solid or another.
I suggest that Infrared spectrum and UV-Vis of Avermectin should be added to manuscript or added to Complementary file.
Response 4: Thanks for your advice, the CuBTC nanomaterials we selected were not purchased, but were synthesized according to the reported literature. For details, please see the preparation of AM@CuBTC. Moreover, Infrared spectrum and UV-Vis of Avermectin was added Complementary file.
UV-Vis spectrum of Avermectin
Infrared spectrum of Avermectin
Results and Discussion:
Point 5: As a whole, all Figures’ legend should be re-written. Figure legends shouldn’t contain acronyms.
In a general way, the manuscript should be written is an impersonal way but in the 398 line appears “we used… In this sense, an impersonal form is close to Avian DF-1 cells were used….
In Fig 2 all hkl planes should be inserted. The X ray diffraction has several diffraction lines but not peaks. Each diffraction line is indexed at one hkl plane. At moment, this figure is very small and should be increased. At moment, there is a lack of information at about adsorption of Avermectin in the surface of nanoparticle, is unclear if this compounds penetrate at surface region. I suggest that infrared at mid region be carried out at least CuBTC and AM@CuBTC.
Figure 3A, 3B, 3C e 3D are very small, and at moment is impossible read bar scale of magnification. The figure should be size increased.
The selection of verbs should be making with extreme careful. In the 455 line, see …was listed in Fig 9. See, the verb to list should be used in a Table.
Response 5: Thanks for your advice, Figure legends have been re-written in the article.
The use of DF-1 cells is simply to prove the uptake efficiency of cells, and to illustrate the cell delivery rule of nano-pesticide. Whether it is human, bird or insect, their cell membrane composition is basically the same, so based on the existing reserves in our laboratory, we chose DF-1 cells, of course we would prefer to use insect cell lines if conditions permit. We are working hard to build better scientific research conditions and research environment.
Figure 2A was the XRD of CuBTC and AM@CuBTC, it can be seen from the X-ray dif-fraction pattern that the peak positions of the characteristic peaks of AM@CuBTC was ba-sically consistent with those of the CuBTC curve, indicating that the purity and crystallin-ity of the sample obtained were close to CuBTC, and Free AM did not change the structure of CuBTC during the process of adsorption.
XRD is reflected in the article to see whether the difference between AM@CuBTC and CuBTC affects the change of structure and the formation of new substances. Therefore, we did a simple characterization, but did not fully analyze the XRD of CuBTC. CuBTC is a nanoparticle that has been reported, so this article will not give a detailed description. CuBTC is a porous structure, and the drug is not distributed on the surface of the material, but in the pores inside the material. The drug-loaded nanomaterials have to be washed with methanol for many times, so there is no possibility of abamectin remaining on the surface, and the drug loading calculated by the difference method is the loading of the drug adsorbed inside the CuBTC. There are many literature reports on CuBTC loading small molecule drugs.
We have adjusted the size of the picture to satisfy readers' viewing. Figure 3 have been size increased according to your suggestion. Moreover, I have removed the verb in Figure 9.
Conclusion:
Point 6: This item is constructed in an erroneous way. Summary is a poor listing of events picked up of Results and Discussion. Then, the item is Conclusion or Summary. If the item is conclusion a set of contributions can be listed. Seems that is viable to write a good conclusion item. See, at moment, this item can be re-written. Sound estrange …“We speculate…”, I suggest insert only one major contributions, and two or three secondary contributions.
Again, I believe that authors make use acronyms in an erroneous way. I does not recommend make use of acronyms in the conclusion item. It is possible, two times a word without problems.
Response 6: Thanks for your advice, we have corrected the Conclusion section based on the suggestion you mentioned, as follows:
In summary, we first proposed a design to load Avermectin through CuBTC on prevention and control pine wood nematode and vector insect Japanese pine sawyer, including evaluation of toxicity mechanism and traceable pesticide monitoring. Utilizing the high porosity and surface area of CuBTC (coordination polymers of copper and trimesic acid) that was chosen as biocompatible material for delivery of Avermectin to improve the solubility, photolysis performance and pesticide efficacy. From traceable monitoring experiment, AM@CuBTC made it easily penetrate the body wall of Japanese pine sawyer larvae and transmit to tissue cells through contact and diffusion. Otherwise, AM@CuBTC can effectively enhance the cytotoxicity and utilization of AM, which will provide valuable research value for the application of typical plant-derived nerve agents in the prevention and control of forestry pests. Moreover, AM@CuBTC could be transmitted to the epidemic wood and dead wood at a low concentration. In the optimal control period of longhorn beetle larvae, the nano-drugs are injected into the parts above the root of the tree by punching the borer, and the nano-drugs are delivered into the interior through the water transport of the tree, and diffuse layer by layer. The release of the hormone, the sustained drug effect will control the larvae, egg hatching and nematode reproduction, which will effectively reduce the incidence of the final disease, as well as effective preventive effect, the slow release of the drug greatly improves the insecticidal efficiency. We speculate that the prepared nano-pesticide will attract lots of focus and enthusiasm in the uptake/inhibition mechanism of other pests, and guide specific target genes, which will break through the limitations of existing research on forestry pests.
Author Contributions:
Point 7: I suggest that complete names and its affiliations be added.
Conceptualization, Y. X. L.; Funding acquisition, C. G. Z.; Methodology, Y. 539 W. Z., X. X., X. Y. X., G. H. W., S. K. G., L. Q. Q., S. Y. Y. and H. X. L.; Resources, C. Y. J., W. X. S. and 540 L. X.; Writing – original draft, Y. X. L.; Writing – review & editing, Y. C. J. All authors have read and 541 agreed to the published version of the manuscript.
Response 7: Thanks for your advice, we have added Author Contributions based on the suggestion you mentioned, as follows:
Author Contributions: Conceptualization, Yanxue Liu; Funding acquisition, Chenggang Zhou; Methodology, Yiwu Zhang, Xin Xin, Xueying Xu, Gehui Wang, Shangkun Gao, Luqin Qiao, Shuyan Yin and Huixiang Liu; Resources, Chunyan Jia, Weixing Shen. and Li Xu; Writing – original draft, Yanxue Liu; Writing – review & editing, Yingchao Ji. All authors have read and agreed to the published version of the manuscript.

Reviewer 3 Report
In this manuscript, the authors reported nano-pesticide prepared by loading AM with MOF-based material CuBTC. Such nanoagent resulted in overall improved pesticide properties such as better biocompatibility, solubility, anti-photolysis and killing efficacy. The material design is interesting, whereas the manuscript could be further polished to make it clearer and more readable. Below are several questions that might be helpful to authors to refine the manuscript.
- Overall this manuscript needs grammar check. Here are several examples: that different from…; with chemicals.6…; there were fluorescence distributed…; as a control in brightfiel…, etc. Also, tenses should be kept consistent.
- Full name of CuBTC should be given, if possible.
- Scale bars in microscopic images in figure 6 should be added.
- Wavelength and power density/output of ultraviolet light used in photodegradability test should be given.
- Nano-pesticide is an attractive research topic and it is gaining increasing attention. The authors should consider talking more about nano-pesticide and adding more relevant citations in Introduction.
Author Response
Response to Reviewer 3 Comments
In this manuscript, the authors reported nano-pesticide prepared by loading AM with MOF-based material CuBTC. Such nanoagent resulted in overall improved pesticide properties such as better biocompatibility, solubility, anti-photolysis and killing efficacy. The material design is interesting, whereas the manuscript could be further polished to make it clearer and more readable. Below are several questions that might be helpful to authors to refine the manuscript.
Point 1: Overall this manuscript needs grammar check. Here are several examples: that different from…; with chemicals.6…; there were fluorescence distributed…; as a control in brightfiel…, etc. Also, tenses should be kept consistent.
Response 1: Thanks for your opinions, I made the following corrections:
1) Utilizing the high porosity and surface area of CuBTC (coordination polymers of copper and trimesic acid) that was chosen as biocompatible material for delivery of AM to improve the solubility, photolysis performance and pesticide efficacy.
2) At present, the prevention and control measures of PWD is mainly through chemical agents control of vector insects.6
3) There was fluorescence distributed on the entire epidermal structure of JPS larvae, especially in the folds, where the fluorescence was intensified, meaning that CuBTC has high adhesion properties.
4) It was clearly observed that the morphology of JPS larvae was successfully labeled with fluorescence under the dark field excitation light of 488 nm, unlabeled CuBTC-treated JPS larvae served as controls (Fig. 6AB).
Point 2: Full name of CuBTC should be given, if possible.
Response 2: Thanks for your advice, the full name of CuBTC is coordination polymers of copper and trimesic acid.
Point 3: Scale bars in microscopic images in figure 6 should be added.
Response 3: Thanks for your advice, I have added scale bars in microscopic images in figure 6. As follows:
Point 4: Wavelength and power density/output of ultraviolet light used in photodegradability test should be given.
Response 4: Thanks for your advice, wavelength and power density/output of ultraviolet light used in photodegradability test was given in Fig. 5.
The retention rate of Free AM, AM@ES, AM@CuBTC with pH of 5.0, 7.0 and 9.0 after being exposed to ultraviolet light (emitted by a 30 W, 310 nm lamp).
Point 5: Nano-pesticide is an attractive research topic and it is gaining increasing attention. The authors should consider talking more about nano-pesticide and adding more relevant citations in Introduction.
Response 5: Thanks for your advice, I have added more relevant citations in Introduction.

Reviewer 4 Report
In this manuscript, authors described the toxicity of Avermectin@CuBTC nanoparticles on vector insect and internal distribution of nanoparticles in insect and tree. This study is very valuable in developing prevention technology for pine wilt disease which is a serious problem with no reliable control method. Here are comments.
- Authors provided valuable in vivo data showing particle distribution in JPS larva and pine tree. Study is focused to control vector insect transmitting disease causing nematodes. It would be better to discuss or provide information on how effective this nanoparticle is to reduce the final disease incidence and inactivate nematodes.
- Authors showed the photolysis of AM. However, this may not be effective when nanoparticles are inside tree because light cannot reach inside tree. What is authors’ opinion?
- If nanoparticles accumulate inside tree, this may affect the industrial quality of pine tree or the vitality of pine tree. What is authors’ opinion?
Author Response
Response to Reviewer 4 Comments
In this manuscript, authors described the toxicity of Avermectin@CuBTC nanoparticles on vector insect and internal distribution of nanoparticles in insect and tree. This study is very valuable in developing prevention technology for pine wilt disease which is a serious problem with no reliable control method. Here are comments.
Point 1: Authors provided valuable in vivo data showing particle distribution in JPS larva and pine tree. Study is focused to control vector insect transmitting disease causing nematodes. It would be better to discuss or provide information on how effective this nanoparticle is to reduce the final disease incidence and inactivate nematodes.
Response 1: Thanks for your advice,I have added a discussion about this part in the conclusion, as follows:
In the optimal control period of longhorn beetle larvae, the nano-drugs are injected into the parts above the root of the tree by punching the borer, and the nano-drugs are delivered into the interior through the water transport of the tree, and diffuse layer by layer. The release of the hormone, the sustained drug effect will control the larvae, egg hatching and nematode reproduction (the inhibition of nematodes will be reflected in the next work), which will effectively reduce the incidence of the final disease, as well as effective preventive effect, the slow release of the drug greatly improves the insecticidal efficiency.
Point 2: Authors showed the photolysis of AM. However, this may not be effective when nanoparticles are inside tree because light cannot reach inside tree. What is authors’ opinion?
Response 2: Thanks for your advice. Abamectin is a broad-spectrum pesticide that has a killing effect on a variety of pests. In addition to being used on longhorn beetles, it still has an effect on the control of other pests. The pests that live on leaves need leaf spraying, so they need to do light performance. Research. One of the reasons for our photolysis experiment is to meet the needs of the general public. On the other hand, we use the method of punching and injecting the nano-pesticide into the tree. The process of injecting the drug also needs to consider the photolysis of the drug. For example, if there is no brown injection bottle, a transparent injection bottle can also be used. The last reason is that we did not avoid light in our indoor research experiments, and we also need to consider its photolysis problem.
Point 3: If nanoparticles accumulate inside tree, this may affect the industrial quality of pine tree or the vitality of pine tree. What is authors’ opinion?
Response 3: From our in vivo imaging experiments, it can be seen that the nanoparticles are uniformly distributed in the tree along with the water transport of the tree, and will not cause severe aggregation. In addition, CuBTC is biodegradable and finally exists in the form of ions or molecules, so it will not be in the form of ions or molecules. Accumulation in the tree body, and the quality of our usage relative to the number is very low, generally does not affect the quality of the tree, in order to thank you for mentioning the problem, we will carry out research in this area. Thank you for your valuable comments.

Round 2
Reviewer 2 Report
Dear Authors:
In my viwpoint, the manuscript "High Adhesion and Penetration of Avermectin@CuBTC Nano-Pesticides for Prevention and Control Pine Wilt Disease, and in vivo Traceable Monitoring" can be accepted to publication after minor revision. This ranking can be justified as follow:
In the Abstract item:
Consider the phrase below. I request that delete this phrase. Really, the verb “to speculate” cannot be used in the abstract
“We speculate that AM@CuBTC will break through the existing research limitations and bring new opportunity to research forestry pests.”
In the Conclusion item:
The item Conclusion exist to present major conclusion and its secondary conclusion. The word conclusion is not a synonymous of Summary. Also, Does not exhibit similar significance.
Therefore, I request that author cut off the word summarizing.
Also, the phrase below:
“We speculate that the prepared nano-pesticide will attract lots of focus and enthusiasm in the uptake/inhibition mechanism of other pests, and guide specific target genes, which will break through the limitations of existing research on forestry pests.”
It is obvious that this phrase is not a conclusion. I suggest that this phrase should be deleted since any contribution or conclusion can be derived of them.
Also, this phrase is written in third person “We” that insert a deleterious characteristic to information since induce an idea of talk with author via its testimonial.

Author Response
Response to Reviewer 2 Comments
In my viwpoint, the manuscript "High Adhesion and Penetration of Avermectin@CuBTC Nano-Pesticides for Prevention and Control Pine Wilt Disease, and in vivo Traceable Monitoring" can be accepted to publication after minor revision. This ranking can be justified as follow:
In the Abstract item:
Point 1: Consider the phrase below. I request that delete this phrase. Really, the verb “to speculate” cannot be used in the abstract
“We speculate that AM@CuBTC will break through the existing research limitations and bring new opportunity to research forestry pests.”
Response 1: Your suggestion is greatly appreciated. We have deleted this phrase.
In the Conclusion item:
Point 2: The item Conclusion exist to present major conclusion and its secondary conclusion. The word conclusion is not a synonymous of Summary. Also, Does not exhibit similar significance.
Therefore, I request that author cut off the word summarizing.
Response 2: Thank you very much for your comments. We have deleted it.
Also, the phrase below:
Point 3: “We speculate that the prepared nano-pesticide will attract lots of focus and enthusiasm in the uptake/inhibition mechanism of other pests, and guide specific target genes, which will break through the limitations of existing research on forestry pests.”
It is obvious that this phrase is not a conclusion. I suggest that this phrase should be deleted since any contribution or conclusion can be derived of them.
Also, this phrase is written in third person “We” that insert a deleterious characteristic to information since induce an idea of talk with author via its testimonial.
Response 3: Your suggestion is greatly appreciated. We have deleted this phrase.
